# Phytochemical Composition of Lichen *Parmotrema hypoleucinum* (J. Steiner) Hale from Algeria

**DOI:** 10.3390/molecules27165229

**Published:** 2022-08-16

**Authors:** Marwa Kerboua, Ali Ahmed Monia, Nsevolo Samba, Lúcia Silva, Cesar Raposo, David Díez, Jesus Miguel Rodilla

**Affiliations:** 1Laboratory of Vegetal Biology and Environment, Biology Department, Badji Mokhtar University, Annaba 23000, Algeria; 2Chemistry Department, University of Beira Interior, 6201-001 Covilhã, Portugal; 3Department of Clinical Analysis and Public Health, University Kimpa Vita, Uige 77, Angola; 4Fiber Materials and Environmental Technologies (FibEnTech), University of Beira Interior, 6201-001 Covilhã, Portugal; 5Mass Spectrometry Service, University of Salamanca, 37007 Salamanca, Spain; 6Department of Organic Chemistry, Faculty of Chemical Sciences, University of Salamanca, 37008 Salamanca, Spain

**Keywords:** lichen, *Parmotrema hypoleucinum*, LC-MSD-Trap-XCT, phytochemical composition, norstictic acid and stictic acid

## Abstract

In this work, we carried out studies of the chemical composition of hexane, chloroform and ethanol extracts from two samples of the lichen *Parmotrema hypoleucinum* collected in Algeria. Each sample of the lichen *P. hypoleucinum* was collected on two different supports: *Olea europaea* and *Quercus coccifera*. Hexane extracts were prepared, in Soxhlet; each hexane extract was fractionated by its solubility in methanol; the products soluble in methanol were separated (cold): 1-Hexane, 2-Hexane; and the products insoluble in methanol (cold): 1-Cires, 2-Cires. A diazomethane esterified sample of 1-Hexane, 2-Hexane, 1-Cires and 2-Cires was analyzed by GC-MS, and the components were identified as methyl esters. In the 1-Hexane and 2-Hexane fractions, the methyl esters of the predominant fatty acids in the lichen were identified: palmitic acid, linoleic acid, oleic acid and stearic acid; a hydrocarbon was also identified: 13-methyl-17-norkaur-15-ene and several derivatives of orsellinic acid. In the 1-Cires and 2-Cires fractions, the previous fatty acids were no longer observed, and only the derivatives of orsellinic acid were found. The analysis of the 1-Hexane, 2-Hexane fractions by HPLC-MS/MS allows us to identify different chemical components, and the most characteristic products of the lichen were identified, such as Atranol, Chloroatranol, Atranorin and Chloroatranorin. In the fractions of 1-Cires and 2-Cires, the HPLC-MS/MS analysis reveals that they are very similar in their chemical components; the characteristic products of this lichen in this fraction are Atranorin and Chloroatranorin. In the extracts of chloroform, 1-Chloroform and 2-Chloroform, the analysis carried out by HPLC-MS/MS shows small differences in their chemical composition at the level of secondary products; among the products to be highlighted for this work, we have chloroatranorin, the stictic acid, norstictic acid and other derivatives. In the analysis of the most polar extracts carried out in ethanol: 1-Ethanol and 2-Ethanol, HPLC-MS/MS analysis shows very similar chemical compositions in these two extracts with small differences. In these extracts, the following acids were identified as characteristic compounds of this lichen: constictic acid, stictic acid, substictic acid and methylstictic acid. In the HPLC–MS/MS analysis of all these extracts, alectoronic acid was not found.

## 1. Introduction

Lichens live in symbiotic associations between fungi and algae and/or cyanobacteria, and in addition to these two symbiotic partners (photobiont and mycobiont) classically described, a third partner can also be integrated: epi and/or endophytic fungi as well as bacteriobiont or associated bacterial communities [1] which are important constituents of many of them. The production of various unique extracellular secondary metabolites known as lichen substances is the result of this symbiosis. The specific condition in which lichens live is the reason for the production of many metabolites that provide good protection against negative physical and biological influences. [2] The majority of lichens-forming fungi belong to Ascomycetes Lecanoromycetes [3]. Due to the vast genetic diversity and interactions with various environmental factors, lichens have unique profiles of primary and secondary metabolites (i.e., lichen substances) with interesting physiochemical properties [4].

In this paper, we describe our studies on the chemical composition of the extracts of *Parmotrema hypoleucinum*, Lecanorales Order, family Parmeliaceae, which, with 2765 species all over the world, is the largest family [5].

In Algeria, the Parmeliaceae family is very present compared to other families [6]; four *Parmotrema* have been identified so far: *P. perlatum*, *P. reticulatum*, *P. robustrum* and *P. hypoleucinum*. The last one is very common in the Mediterranean area [7]; it belongs to the Lecanorales Order and to the Parmeliaceae Family. It corresponds to a foliaceous lichen that can be up to 12 cm in diameter. In general, the lobes are very irregular, wide and raised, often forming tufts on small branches and can appear like curly lettuce. The upper side has a grayish appearance, and the lower side is largely white, which allows it to be easily distinguished from other similar species that have a dark back side. This species has black cilia and marginal Soralies.

Several activities of these lichen molecules, mainly those resulting from the polymalonate acetate pathway, are of interest for cosmetics: photoabsorbing, antioxidant and inducing melanogenesis. These properties have been studied for a still limited number of secondary metabolites (Mitrović, 2011) (Figure 1).

The natural products isolated from different lichens (such as Usnic acid, Lobaric acid, Atranorin, Protolichesterinic acid and Salazinic acid) have good antibiotic activities against Gram-positive bacteria and are also active against pathogenic dermatophyte fungi [8]. Other products found in lichens, such as anthraquinones derivatives, bianthrones and hypericin, have an inhibitory action on the activity of viral enzymes, such as the integrase of HIV-1 and HSV-1 [9,10] and also on enzymes such as lipooxygenases, histidine decarboxylase and tyrosinase; other derivatives inhibit the biosynthesis of Leukotriene B_4_ (LTB_4_) [11,12,13].

Polyphenolic products isolated from lichens have limitations, low solubility and, above all, toxicity. Usnic acid is a polyphenolic compound very common in lichens that has good activity, among others, against microorganisms of the Mycobacterium genus. In the 1950s, Shibata and Miura [14] made modifications and derivatizations of the functional groups of usnic acid to carry out the structure–activity correlation study to enhance the biological activity profile.

Usnic and polyporic acids have a good growth inhibition activity of L1210 leukemic cells; in subsequent research [15,16], several derivatives of these acids have been prepared to enhance antitumor activity, but none of these derivatives have exceeded the activities presented by usnic and polyporic acids.

Kumar and Muller have prepared a series of analogs of barbatic, diffractaic and obtusatic acids isolated from lichens to evaluate the effects of inhibition of the biosynthesis of LTB_4_ and as antiproliferative agents. Some of these derivatives show good potential as LTB_4_ biosynthesis inhibitors [11,13]. 

Lichens can also have xanthones, that shown enzyme modulation that are therapeutic targets, such as protein kinase C [17], topoisomerase II [18,19], acetylcholinesterase [20] and monoamine oxidases [21]; antiretrovirals [22,23], antimalarials [24,25], antihypertensives [26], anti-inflammatory, cytotoxics [27] and antitumors [28,29]. For this reason, the use of the base skeleton of xanthone is justified to prepare derivatives with bioactive potential.

Many depsidones isolated from lichens and higher plants have important activities, including the inhibition of enzymatic activity [30], antimycobacterial, anti-inflammatory, analgesic, antitumor, cytotoxic and antiviral activity [31,32,33]. In the work published in 2015 by James C Lendemer and collaborators [34], they studied and delineated the *Parmotrema* species in eastern North America. Using morphological, chemical, reproductive and ecological characters, they define four species for this group: *P. hypoleucinum*, *P. hypotropum*, *P. perforatum* and *P. subrigidum*.

This group has found *P. hypoleucinum* and *P. subrigidum* to be momophyletic, the latter comprising two chemotypes that differ in the presence or absence of norstictic acid in addition to alectoronic acid.

Due to the pharmacological potential presented by compounds isolated in lichens, it was decided to study the chemical composition of the lichen *Parmotrema hypoleucinum* (J. Steiner) Hale, collected on two different supports in the area of Lac Tonga in Algeria.

## 2. Results and Discussion

*Parmotrema hypoleucinum* (J. Steiner) Hale is an epiphytic lichen collected in Algeria and studied in order to determine its metabolic composition and chemical fingerprint. *P hypoleucinum* (J. Steiner) Hale was collected in two different supports, the first one in Lac Tonga (Sector Brabtia) on *Olea europaea* and the second one in Lac Tonga (Sector Brabtia) on *Quercus coccifera*. The metabolic compositions of each lichen sample were studied by sequential extraction, first of all, with hexane in Soxhlet for low polarity products. The remaining vegetable mass was placed with chloroform at room temperature to obtain the chloroform extract for the products of intermediate polarity, and finally, the vegetable mass was extracted with ethanol at room temperature for the products with higher polarity.

The hexane extract of each sample was dissolved in hot methanol and allowed to cool slowly to obtain the products insoluble in cold methanol. In this way, the products insoluble in methanol were separated: **1-Cires** and **2-Cires**; remaining soluble: **1-Hexane** and **2-Hexane**. Initially, an aliquot of the samples **1-Hexane**, **2-Hexane, 1-Cires** and **2-Cires** were esterified with diazomethane to esterify the acid groups of the existing compounds. These esterified samples were analyzed by GC-MS to identify compounds of lower polarity. 

The different extracts obtained were:

From *P hypoleucinum* (J. Steiner) Hale on *Olea europaea*
**1-Hexane, 1-Cires, 1-C 1-Ethanol**
From *Parmotrema hypoleucinum* (J. Steiner) Hale on *Quercus coccifera*.
**2-Hexane, 2-Cires, 2-Choroform, 2-Ethanol**


Chemical analysis of the hexane extract soluble in MeOH esterified with diazometane, 1-Hexan and 2-Hexane of Parmotrema hypoleucinum collected from two different phorophytes by GC/MS

The **1-Hexane** and **2-Hexane** samples were esterified with diazomethane to esterify the existing acid groups to their methyl esters for GC-MS analysis, being the natural products of the acids indicated in Table 1 for **1-Hexane** sample and Table 2 for **2-Hexane** sample.

The **1-Hexane** sample was analyzed by GC-MS, and eight products were identified, among them palmitic, linoleic, oleic and stearic acids and an unidentified compound. Figure 2 and Table 1. The esterified **2-Hexane** sample was also analyzed by GC-MS, identifying six products, palmitic, linoleic, oleic and stearic acid, a phenolic compound 2,4-dihydroxy-3,5,6-trimethylbenzoic acid and 13-methyl-17-norkaur-15-ene. Figure 3 and Table 2.

The compound identified as Hibaene (13-methyl-17-norkaur-15-ene) is the product with a retention time of 23:53 in GC-MS; its mass spectrum shows the molecular ion at 272 and comes from the dehydration of alcohol (-)-*ent*-Kauran-16*α*-ol in the ionization source of the mass spectrometer. The natural product should be the alcohol (-)-*ent*-Kauran-16*α*-ol.

*P hypoleucinum* (J. Steiner) Hale, on *Quercus coccifera*, the **2-****Hexane** sample, it has a lower number of components, and all were identified in the **1-Hexane** sample. The most important fatty acids in the extracts were identified as palmitic, linoleic, oleic and stearic acids. 2,4-Dihydroxy-3,5,6-trimethylbenzoic acid and (-)-*ent*-Kauran-16*α*-ol alcohol is also identified in the two samples, **1-Hexane** and **2-Hexane**. Only in the **1-Hexane** sample 4-hydroxy-2-methoxy-3,5,6-trimethylbenzoic acid and 2,4-dihydroxy-3,6-dimethylbenzoic acid are also identified. 

For the fractions that have been obtained by crystallization from the crude hexane extract by solubilization in hot methanol, **1-Cires** and **2-Cires** are also esterified with diazomethane for GC-MS analysis.

In the **1-Cires** analysis, five products are found, with four being identified. Figure 4 and Table 3.

In the **2-Cires** sample, three products have been identified from the methanol-insoluble part of the hexane extract of *P. hypoleucinium* (*Quercus coccifera*). Figure 5 and Table 4.

In the **1-Cires** sample, there is a compound that could not be identified; this product does not appear in the **2-Cires** sample. The kaurene-skeletal alcohol is found in **1-Cires** and was not found in the **2-Cires** sample. Of the other three compounds, two were identified in **1-Cires** and **2-Cires**: 2,4-dihydroxy-3,5,6-trimetylbenzoic acid and 2,4-dihydroxy-3,6-dimethylbenzoic acid.

In the **1-Cires** sample, 4-hydroxy-2-methoxy-3,6-dimethylbenzoic acid is also identified, and this product does not appear in the 2-Cires sample; instead, the 2,4-dimethoxy-6-methylbenzoic acid appears in **2-Cires**.

In the HPLC—MS/MS analysis of the **1-Hexane** and **2-Hexane** fractions of *P. hypoleucinium*, 95 products were detected in the **1-Hexane** fraction, in which we proposed 78 structures (and 16 not identified). In total, 91 products were found in the **2-Hexane** fraction, of which we proposed 78 structures (and 13 no identified ones). 

Figure 6 and Figure 7 show the HPLC chromatograms that allowed this analysis and identification of the indicated compounds to be carried out; the complete result is shown in Table 5.

The most characteristic products of the **1-Hexane** and **2-Hexane** extracts of the lichens in the two samples are 3-methylorsellinic acid, atranol, 4-formylbenzoic acid, chloroatranol, *p*-coumaric acid, atranorin and chloroatranorin.

In the fatty acid profile, we find the following acids common to the two samples: 2-hydroxy-10-undecenoic, 2-undecenedioic, undecanedioic, trans-2-dodecenedioic, 9-hydroxy-10,12-pentadecadienoic, tridecanedioic, 9Z-octadecenedioic, 10-acyl-9-formyl-13-hydroxyoctadeca-6,11-dienoic, octadecanedioic, 2-oxopalmitic, nonadecanoic, arachidic and 12-triacontenedioic acids.

Among the acids that have a cyclohexanecarboxylic base, the following derivatives were also found, as described in the publication of the lichen *Physcia mediterranea* Nimis [35], the 3,5-dimethoxycyclohexanecarboxylic acids, 6-(hydroxymethyl)-3,5-bis(methoxycarbonyl)-2,4-dimethylcyclohex-1-ene-1-carboxylic, 3,5,6-hydroxymethyl-2,4-dimethylcyclohex-1-ene-1-carboxylic, 5-formyl-3-hydroxymethyl-2,4,6-trimethylcyclohex-1-ene-1-carboxylic, 3,5-dihydroxy-2,4,6-trimethylcyclohex-1-ene-1-carboxylic, 5-formyl-3,6-dihydroxymethyl-2,4-dimethylcyclohex-1-ene-1-carboxylic, 2,4-dihydroxy-3,5,6-trimethylcyclohexane-1-carboxylic, 4-hydroxy-2,5-dimethylcyclohex-1-ene-1-carboxylic, 6-(1-oxopentyl)-cyclohex-1-ene-1-carboxylic acids.

Lactones were also identified: *N*-dodecanoyl-*L*-homoserine lactone and fukinanolide, other derivatives such as leoidin, lecideoidin, an alcohol such as 7,10,12-nonadecatrien-1-ol and a triterpenic acid identified as ursolic acid (found in the two extracts).

In the comparison of the characteristic polyphenolic compounds of the lichens in these two samples, 20 compounds were identified: 4-O-demethyldivaricatic, barbatic, 8-hydroxybarbatic, baeomycesic, allo-protolichesterinic, 4′-O-demethylsekikaic acids, 8-hydroxydiffractaic, glomelliferonic occurs in *Xanthoparmelia subincerta* [36], muronic is detected in *P. praesorediosum* [37], murolic, lichesterylic [35], praesorediosic, 19-acetoxyprotolichesterinic, diploicin, leprolomin, methyl 3-formyl-2-hydroxy-4-((4-methoxy-2-methylbenzoyl)oxy)-6-methylbenzoate, superpicrolichenic and 3-formyl-2-hydroxy-4-((2-(14-hydroxypentadecyl)-4-methyl-5-oxo-2,5-dihydrofuran-3-carbonyl)oxy)-6-methylbenzoic acids. 

In the **1-Hexane** sample, the following compounds were not identified: 3,5-dimethylorselinic acid, 9,10-dihydroxy-8-oxo-12-octadecenoic acid, octadecanedioic acid and 18-hydroxylinoleic acid, which were found in the **2-Hexane** sample. The following compounds were not identified in the **2-Hexane** sample: barbatic acid, tetraoxodocosanoic acid and superpicrolichenic acid; these products were identified in the **1-Hexane** sample too. In the 1-Hexane sample, there are 17 products that could not be identified and in the **2-Hexane** 13.

In the analysis of the **1-Cires** and **2-Cires** fractions, obtained by precipitation of the initial hexane extract by HPLC-MS/MS, it has been possible to detect in **1-Cires** 56 products, of which 11 products were not identified. In **2-Cires**, 53 products were detected, and 13 products could not be identified. For the acids and diacids identified in **1-Cires** there are 23 products, and in **2-Cires**, we have 27 products. As benzoic acids or derivatives we have p-coumaric acid and 6,7-dihydroxycoumarin. Among the polyphenolic compounds and esters, **1-Cires** and **2-Cires** were identified: *allo*-protolichestrinic acid, atranorin, 7-chloro-3-oxo-1,3-dihydroisobenzofuran-5-carboxylic acid, chloroatranorin, 8-hydroxydiffractaic acid and 19-acetoxylichestrinic acid. 

The chromatograms and the analysis of the **1-Cires** and **2-Cires** compounds are shown in Figure 8 and Figure 9 and Table 6.

In the work carried out in 2016, in a general analysis of the chemical relationship in the group of *Parmotrema perforatum* (Parmeliaceae, Ascomycota), each sorediate species is descended from an apotheciate species with the same secondary chemicals [38]. The lichen *Parmotrema hypoleucinum* is derived from the ancestor *Parmotrema perforatum* represented by its secondary metabolites in stictic, constictic and norstictic acids. These acids have not been extracted in the Hexane extract, and they do not exist in the part insoluble in methanol, **1-Cires** and **2-Cires** nor in the part soluble in methanol: **1-Hexane** and **2-Hexane**.

The products identified in the **1-Chloroform** and **2-Chloroform** extracts are shown in the chromatograms in Figure 10 and Figure 11 and the results of the identification of the compounds in Table 7.

In the analysis of **1-Chloroform** and **2-Chloroform** extracts, the acids of the secondary metabolites that define the Sorediate species are identified for *Parmotrema hypoleucinum*; these acids are as follows. Constictic acid is also detected in *Parmotrema tinctorum*; Norstictic acid, according to mycologia 2015, this compound appeared to be present in variable concentrations throughout the thallus. Often the medulla of a lobe tested negative while the medulla adjacent to the apothecia tested positive or vice versa. Stictic acid and other derivatives identified as Substictic acid, there are depsidone were detected in *Parmotrema tinctorum, P. grayanum*, also in *P. robustum* and *P. andinum* [39].

In sample **1-Chloroform**, 60 products were detected, of which 52 products were identified, and 8 products were not identified, and in the sample **2-Chloroform**, 59 compounds were detected, of which 52 products were identified and 7 unidentified.

As in the previous analyses, fatty acids, hydroxy acids, oxo acids, some xanthones and flavones have been found, in addition to menegazziaic, siphullelic, protocetraric, conphysodalic, cryptostictic, menegazziaic isomer, gyrophoric, lecanoric and muronic acids among others.

In the extracts of the more polar products made with ethanol, in the **1-Ethanol** sample 57 products were found, of which 54 products were identified, and 3 compounds were not identified. Figure 12 and Table 8. In the sample **2-Ethanol** analyzed, 54 compounds were detected, of which 52 products were identified, and 2 compounds were not identified. Figure 13 and Table 8.

Among the compounds identified in the 1-Ethanol and 2-Ethanol samples were also found the acids that define this lichen *Parmotrema hypoleucinum*, the Constictic and Stictic acids and a derivative such as methylstictic acid.

The products analyzed by GC-MS were identified by their mass spectra and compared with the mass spectra of the NIST and Wiley databases.

## 3. Materials and Methods

### 3.1. Lichen Material

*Parmotrema hypoleucinum* is a foliose epiphytic lichen, which was collected on *Quercus coccifera* and on *Olea europea* at Lake Tonga (Sector Brabtia), at an altitude of 2.20 m above sea level, coordinate 36°51′38″ N; 08°28′46″ E in June 2017. The area of lake tonga is 2600 ha communicating with the sea through the artificial channel of the Messida.

This station is located in the national park of el kala (80,000 ha) Figure 14, classified as a biosphere reserve by UNESCO in 1990, located in the extreme northeast of Algeria.

*Parmotrema hypoleucinum* (J. Steiner) Hale was identified by Professor Monia Ali Ahmed lichenologist and research director of the Pathology of Ecosystems team at the University of Badji-Mokhtar, Annaba, Algeria. This sample has been deposited in Badji-Mokhtar University, Annaba, code AAM-2.

### 3.2. HPLC Orbitrap

#### 3.2.1. Sample Preparation

Hexane extraction was carried out for each lichen sample; for *Parmotrema hypoleucinium* (J. Steiner), Hale was made from 40 g of powder material for the two samples. The extraction was carried out in Soxhlet apparatus with hot n-Hexane for 24 h; after this time, the solvent was evaporated to obtain the *Parmotrema hypoleucinium* (*Olea europaea*) n-Hexane extract 0.66 g, which represents (1.65%) and *Parmotrema hypoleucinium* (*Quercus coccifera*)) n-Hexane extract 0.27 g, which represent (0.68%). The n-Hexane extracts are then dissolved in hot methanol and allowed to cool to room temperature so that insoluble products crystallize. With this treatment, the methanol insoluble part (cires) and the cold methanol soluble part (Hexane) are obtained for each n-Hexane extract. For *P hypoleucinium,* the part insoluble in methanol produced **1-Cires** of 0.379 g and **2-Cires** of 0.058 g; the part soluble in methanol produced **1-Hexane** of 0.281 g and **2-Hexane** of 0.219 g.

The vegetable mass recovered and dried from the extractions with Hexane was placed to extract with chloroform at room temperature for 5 days. After this time, the chloroform extract was filtered, and the solvent was evaporated, obtaining the **1-Chloroform** extract with a weight of 0.226 g, which represents (5.65%) and the **2-Chloroform** extract with a weight of 0.196 g, which represents (4.90%).

After being extracted the vegetable mass with chloroform was placed with ethanol for 5 days to prepare the Ethanol extracts. Evaporation of the solvent gave the ethanol extracts: for **1-Ethanol,** a mass of 2385 g was recovered, which represented 5.96%, and for the **2-Ethanol** extract, it presented a mass of 2.417 g, which represented 6.04%.

#### 3.2.2. Instruments

For the GCMS analysis, an Agilent MS220 mass spectrometer coupled to a 7890A GC was used.

HPLC analyses were carried out on an orbitrap Thermo q-Exactive mass spectrometer coupled to a Vanquish HPLC.

#### 3.2.3. GCMS Parameters

The oven temperature was initially set to 50 °C, held for 5 min and then a ramp of 30 °C/min was applied up to 270 °C that was held for 5 additional mins. A VF-5 ms columns was used, with a length of 30 m, inner diameter 0.25 mm and layer width of 0.25 micron.

MS spectra were acquired in EI mode with a mass range from 50 uma to 600 uma.

#### 3.2.4. LC Parameters

For the HPLC separation, a Kinetex XB-C18 (Phenomenex) with a particle size of 2.6 microns, 100 mm in length and a diameter of 2.1 mm was used as column. As solvent A, water with 0.1% of formic acid was used, and as solvent B, acetonitrile was chosen. The column flow war 0.200 mL/min. The following gradient was used (in Table 9):

#### 3.2.5. MS Parameters

For the ionization electrospray in negative mode was used, with the following parameters: Electrospray voltage −3.8kV, Sheath gas 30, Aux gas 10, drying gas temperature 310 ºC. Capillary temp. 320 and S-lens value of 55.0.

The acquisition was performed in a mass range from 100 to 1000 uma, and an auto MS2 program was used with a fragmentation voltage of 30.

## 4. Conclusions

Due to the biological importance of lichens isolated natural products, studies of the chemical composition of two samples of the lichen *Parmotrema hypoleucinum*, collected on two different supports: Olea europaea and Quercus coccifera, in Algeria, were carried out. For each sample, the extracts of hot *n*-Hexane, Chloroform at room temperature and Ethanol at room temperature were carried out.

The *n*-Hexane extract of each sample is dissolved in hot methanol and allowed to cool slowly so that the products insoluble in methanol precipitate. The parts soluble in methanol are obtained by filtration, the solvent is evaporated, and they are designated as **1-Hexane** and **2-Hexane** fractions for each sample, respectively. The product insoluble in methanol, washed and dried, are designated as the **1-Cires** and **2-Cires** fractions, respectively. An aliquot sample of the fractions: **1-Hexane**, **2-Hexane**, **1-Cires** and **2-Cires**, were esterified with diazomethane to produce the methyl esters of the existing acids.

Esterified samples of **1-Hexane** and **2-Hexane** were analyzed by GC-MS to identify components of lower polarity. In both samples, the methyl esters of 2,4-dihydroxy-3,5,6-trimethylbenzoic acids, palmitic acid, linoleic acid, oleic acid, stearic acid and the hydrocarbon 13-methyl-17-norkaur-15-ene (Probably the natural product will be (-)-ent-kauran-16α-ol). 4-Hydroxy-2-methoxy-3,5,6-trimethylbenzoic and 2,4-dihydroxy-3,6-dimethylbenzoic acid methyl esters and a product that does not it has been possible to identify mass 288; these products were not found in the esterified sample of **2-Hexane**.

The **1-Cires** and **2-Cires** esterified samples were analyzed by GC-MS to separate and identify the components of lower polarity. In the analysis carried out, 2,4-dihydroxy-3,5,6-trimethylbenzoic and 2,4-dihydroxy-3,6-dimethylbenzoic acids were found as existing products in the two samples. In the **2-Cires** esterified sample, 2,4-dimethoxy-6-methylbenzoic acid, which does not exist in the esterified **1-Cires** fraction, was also identified in this analysis. In the esterified fraction **1-Cires**, the analysis also identified 4-hydroxy-2-methoxy-3,6-dimethylbenzoic acid, the hydrocarbon 13-methyl-17-norkaur-15-ene, and a product that was not identified with a mass of 312, which were not found in the esterified sample of **2-Cires**.

The analysis, identification and comparison of the components of the original fractions of **1-Hexane** and **2-Hexane** by HPLC-MS/MS indicate that they are very similar and show almost no differences, except for some very minor components; in these fractions of low polarity, the most characteristic components *P hypoleucinum* will be: Atranol, Chloroatranol, Atranorin and Chloroatranorin; a triterpenic acid identified as Ursolic acid has also been found in both samples.

The analysis of the original **1-Cires** and **2-Cires** fractions by HPLC-MS/MS shows that they are very similar, presenting some small differences in some minor components. The predominant components identified are the fatty acids indicated in the Tables in addition to the predominant product Atranorin and Chloroatranorin, which will be characteristic for this lichen.

In the analysis carried out by HPLC-MS/MS of the original extracts of **1-Chloroform** and **2-Chloroform** of the two samples of *P hypoleucinum* and in the comparison of their components, small differences were found in some secondary compounds. In this analysis, the following acids can be considered as characteristic products of the lichen: Constictic acid, Norstictic acid (this acid was only found in the **1-Chloroform** extract), Stictic acid, Substictic acid and Chloroatranorin.

The HPLC-MS/MS analysis of the original extracts of **1-Ethanol** and **2-Ethanol** did not suggest great differences, and most of their components were identified; the following compounds were found as more representative of the lichen: Constictic acid, Stictic acid, Substictic acid; Methylstictic acid was only identified in the extract of 1-Ethanol.

In this work, a more complete study of the components of the different extracts made from *P. hypoleucinum* was carried out due to the importance of the biological activities. In the most important identified products, in general, their biological activities were referred to derivatives of orsellinic acid, atranol, chloroatranol, atranorin, chloroatranorin, stictic acid and norstictic acid.

For these components, their antioxidant activities, apoptotic effects, cytotoxic, antimicrobial and antitumor activity are already known.

## Figures and Tables

**Figure 1 molecules-27-05229-f001:**
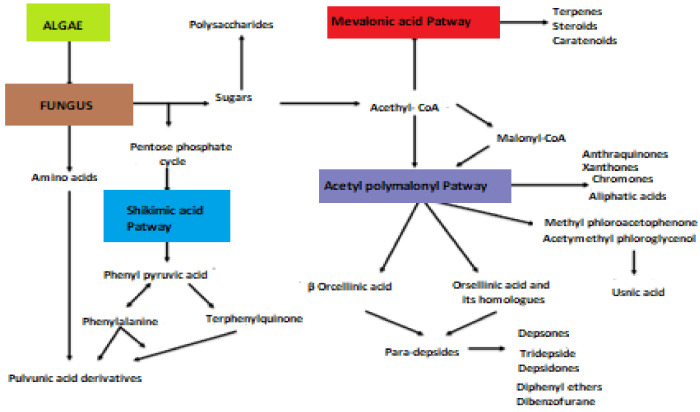
Biosynthetic pathways of lichen secondary metabolites (Elix, 1996; Stocker wörgötter, 2008).

**Figure 2 molecules-27-05229-f002:**
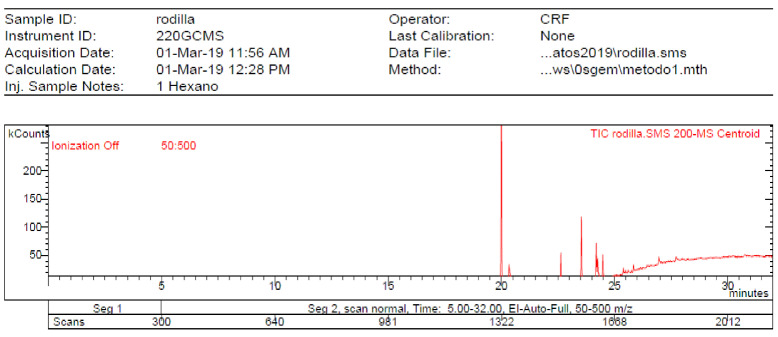
Chromatogram of Hexane extract part soluble in MeOH esterified with diazomethane, **1-Hexane**.

**Figure 3 molecules-27-05229-f003:**
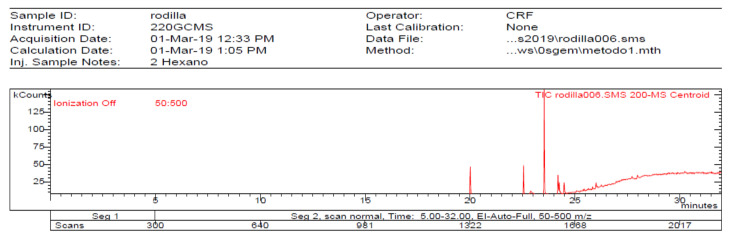
Chromatogram of Hexane extract part soluble in MeOH esterified whit diazomethane, **2-Hexane**.

**Figure 4 molecules-27-05229-f004:**
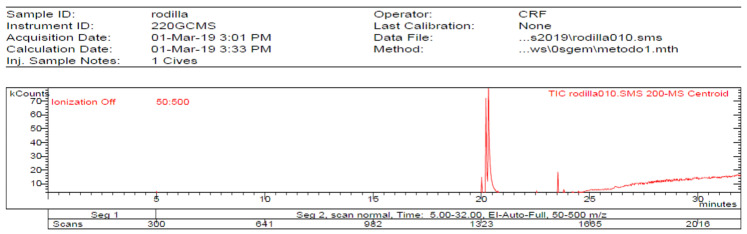
Chromatogram of Hexane extract part insoluble in MeOH esterified whit diazomethane, **1-Cires**.

**Figure 5 molecules-27-05229-f005:**
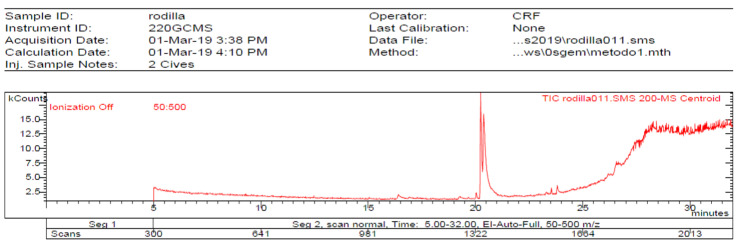
Chromatogram of Hexane extract part insoluble in MeOH esterified whit diazometane, **2-Cires**.

**Figure 6 molecules-27-05229-f006:**
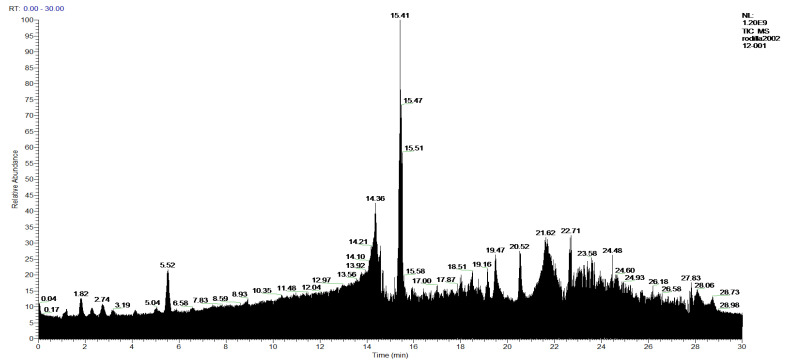
Chromatogram of Hexane extract part soluble in MeOH, **1-Hexane**.

**Figure 7 molecules-27-05229-f007:**
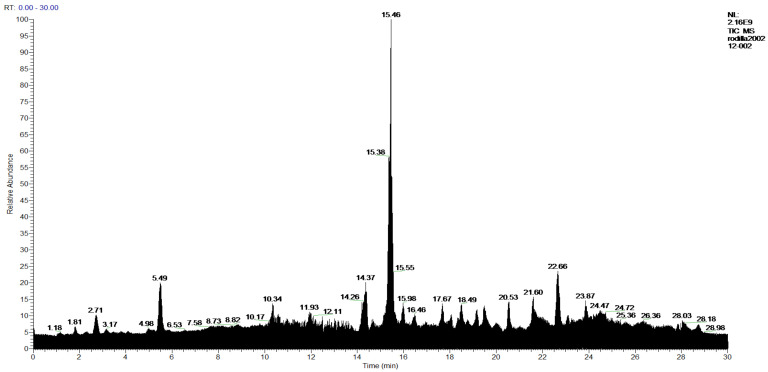
Chromatogram of Hexane extract part soluble in MeOH esterified whit diazomethane, **2-Hexane**.

**Figure 8 molecules-27-05229-f008:**
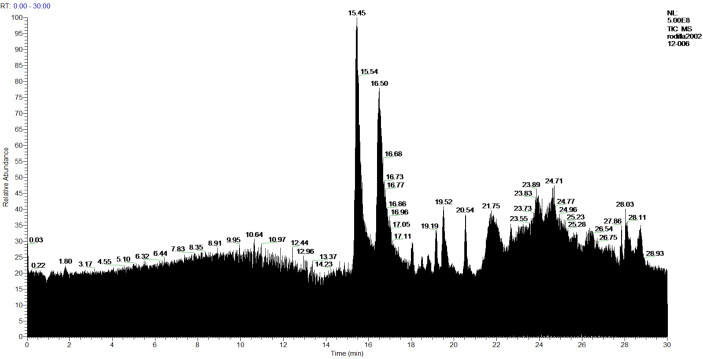
Chromatogram of Hexane extract part insoluble in MeOH, **1-Cires**.

**Figure 9 molecules-27-05229-f009:**
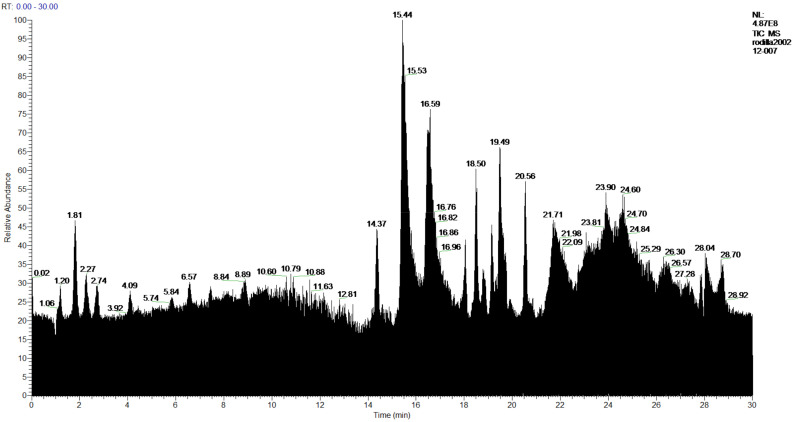
Chromatogram of Hexane extract part insoluble in MeOH, **2-Cires**.

**Figure 10 molecules-27-05229-f010:**
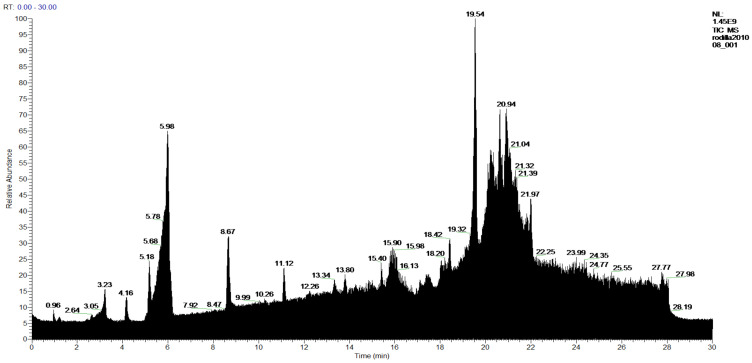
Chromatogram of **1-Chloroform** extract from *P. hypoleucinum*.

**Figure 11 molecules-27-05229-f011:**
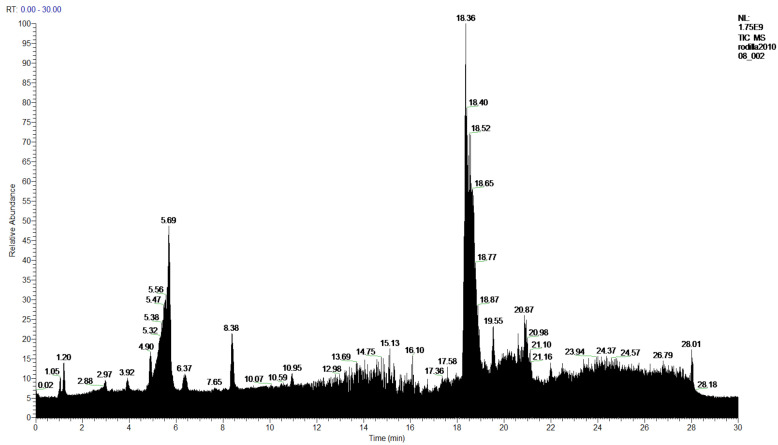
Chromatogram of **2-Chloroform** extract from *P. hypoleucinum*.

**Figure 12 molecules-27-05229-f012:**
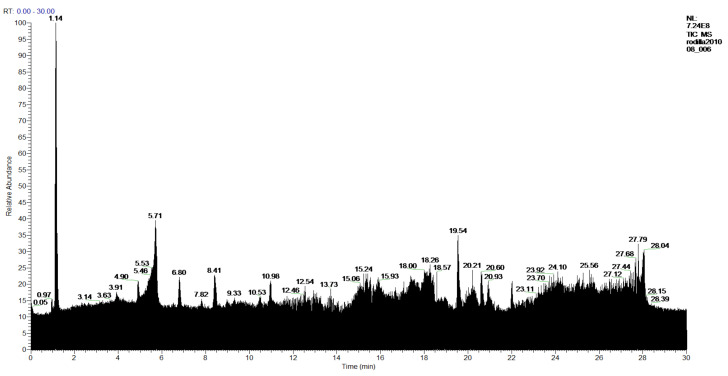
Chromatogram of **1-Ethanol** extract from *P. hypoleucinum*.

**Figure 13 molecules-27-05229-f013:**
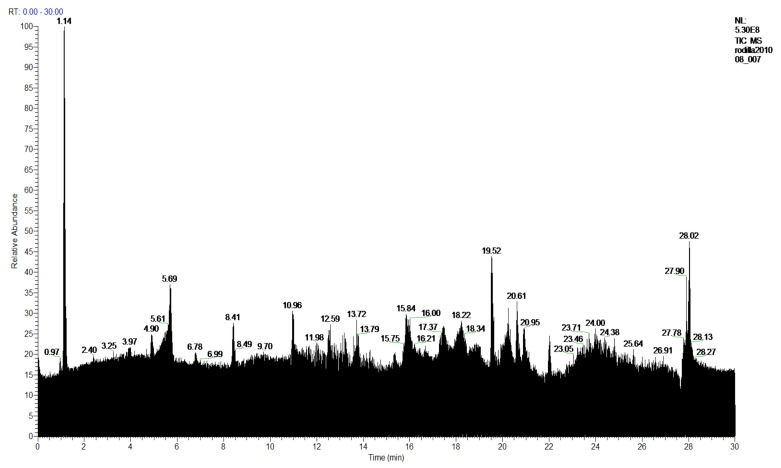
Chromatogram of **2-Ethanol** extract from *P. hypoleucinum*.

**Figure 14 molecules-27-05229-f014:**
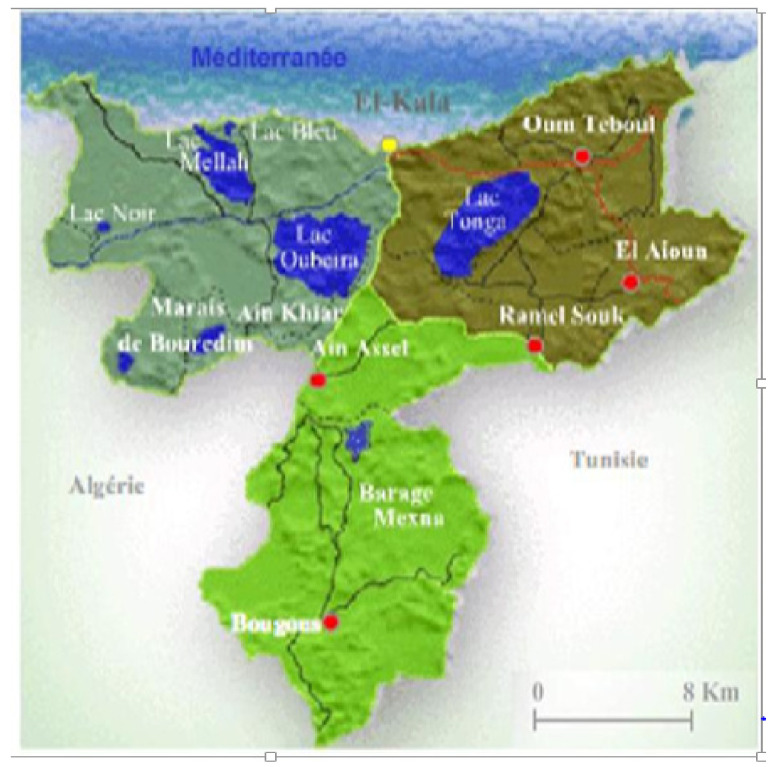
Location of El Kala National Park (P.N.E.K., 2010).

**Table 1 molecules-27-05229-t001:** Sample *Parmotrema hypoleucinum* (in *Olea europaea*), Hexane extract part soluble in MeOH esterified whit diazomethane, **1-Hexane**.

Nº	RT	Identified Product	Mass	%	Natural Compound, Structure
1	19:84	Methyl 4-hydroxy-2-methoxy-3,5,6-trimethylbenzoate	224	0.5	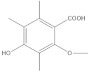
2	20:00	Methyl 2,4-dihydroxy-3,5,6-trimethylbenzoate	210	62.4	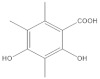
3	20:34	Methyl 2,4-dihydroxy-3,6-dimethylbenzoate	196	8.6	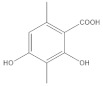
3	22:36	Methylhexadecanoate	270	5.3	C_16_H_32_O_2_ palmitic acid
4	23:53	13-methyl-17-norkaur-15-ene (hibaene)	272	10.1	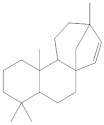
5	24:19	Methyl 9,12-octadecadienoate	294	4.2	C_18_H_32_O_2_ linoleic acid
6	24:25	Methyl (*Z*)-9-octadecenoate	296	2.1	C_18_H_34_O_2_ oleic acid
7	24:48	Methyloctadecanoate	298	3.6	C_18_H_34_O_2_ stearic acid
8	25:39	Unidentified	288	0.8	Unidentified

**Table 2 molecules-27-05229-t002:** Sample *Parmotrema hypoleucinum* (in *Quercus coccifera*), Hexane extract part soluble in MeOH esterified whit diazometane, **2-Hexane**.

Nº	RT	Identified Product	Mass	%	Natural Compound, Structure
1	20:01	Methyl 2,4-dihydroxy-3,5,6-trimethylbenzoate	210	2.5	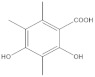
2	22:55	Methylhexadecanoate	270	2.8	C_16_H_32_O_2_ palmitic acid
3	23:54	13-Methyl-17-norkaur-15-ene Probably the natural product will be (-)-*ent*-Kauran-16*α*-ol	272	36.7	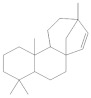
4	24:19	Methyl (*Z,Z*)-9,12-octadecadienoate	294	1.2	C_18_H_32_O_2_ linoleic acid
5	24:23	Methyl (*Z*)-9-octadecenoate	296	1.0	C_18_H_34_O_2_ oleic acid
6	24:38	Methyl octadecanoate	298	1.4	C_18_H_34_O_2_ stearic acid

**Table 3 molecules-27-05229-t003:** Sample *Parmotrema hypoleucinum* (in *Olea europaea*), Hexane extract part insoluble in MeOH esterified whit diazomethane, **1-Cires**.

Nº	RT	Identified Product	Mass	%	Natural Compound, Structure
1	20:02	Methyl 2,4-dihydroxy-3,5,6-trimethylbenzoate	210	1.1	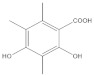
2	20:21	Methyl 4-hydroxy-2-methoxy-3,6-dimethylbenzoate	210	26.6	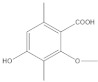
3	20:33	Methyl 2,4-dihydroxy-3,6-dimethylbenzoate	196 C_9_H_10_O_4_	38.9	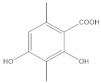
4	22:51	Unidentified	312	0.9	Unidentified
5	22:51	13-Methyl-17-nor-8*β*,13*β*-kaur-15-ene Probably the natural product will be (-)-*ent*-Kauran-16*α*-ol	272 C_20_H_32_ Hibaene	1.8	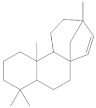

**Table 4 molecules-27-05229-t004:** Sample *Parmotrema hypoleucinum* (in *Quercus coccifera*), Hexane extract part insoluble in MeOH esterified whit diazomethane, **2-Cires**.

Nº	RT	Identified Product	Mass	%	Natural Compound, Structure
1	20:24	Methyl 2,4-dihydroxy-3,5,6-trimethylbenzoate	210	0.9	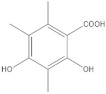
2	20:35	Methyl 2,4-dimethoxy-6-methylbenzoate	210	41.4	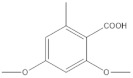
3	20:38	Methyl 2,4-dihydroxy-3,6-dimethylbenzoate	196	48.9	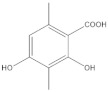

**Table 5 molecules-27-05229-t005:** Samples *Parmotrema hypoleucinum* (in *Olea europea*), Hexane extract part soluble in MeOH **1-Hexane** and *Parmotrema hypoleucinum* (in *Quercus coccifera*), Hexane extract part soluble in MeOH, **2-Hexane**.

Nº	RT	[M-H]^−^	Mass Calc	Formula	Formula	Compounds
				**1-Hexane**	**2-Hexane**	
1	2.67	181.0502	182.0574	C_9_H_10_O_4_	C_9_H_10_O_4_	3-Methylorsellinic acid
2	2.72	187.0970	188.1043	C_9_H_16_O_4_	C_9_H_16_O_4_	3,5-Dimethoxyciclohexanecarboxilic acid
3	2.75	293.0669	294.0741	C_14_H_14_O_7_	C_14_H_14_O_7_	6-(Hydroxymethyl)-3,5-bis(methoxycarbonyl)-2,4-dimethylcyclohex-1-ene-1-carboxylic acid
4	3.08	243.1239	244.1311	C_12_H_20_O_5_	C_12_H_20_O_5_	3,5,6-Hydroxymethyl-2,4-dimethylcyclohex-1-ene-1-carboxylic acid
5	3.15	151.0393	152.0465	C_8_H_8_O_3_	C_8_H_8_O_3_	Atranol
6	3.20	225.1129	226.1201	C_12_H_18_O_4_	C_12_H_18_O_4_	5-Formyl-3-hydroxymethyl-2,4,6-trimethylcyclohex-1-ene-1-carboxylic acid
7	3.47	199.0973	200.1046	C_10_H_16_O_4_	C_10_H_16_O_4_	3,5-Dihydroxy-2,4,6-trimethylciclohexenecarboxilic acid
8	3.55	195.0660	196.0730	-	C_10_H_12_O_4_	3,5-Dimethylorsellinic acid
9	3.58	149.0237	150.0310	C_8_H_6_O_3_	C_8_H_6_O_3_	4-Formylbenzoic acid
10	3.82	241.1081	242.1153	C_12_H_18_O_5_	C_12_H_18_O_5_	5-Formyl-3,6-dihydroxymethyl-2,4-dimethylcyclohex-1-enecarboxylic acid
11	4.10	201.1129	202.1202	C_10_H_18_O_4_	C_10_H_18_O_4_	2,4-Dihydroxy-3,5,6-trimethylcyclohexane-1-carboxylic acid
12	4.95	199.1337	200.1409	C_11_H_20_O_3_	C_11_H_20_O_3_	2-Hydroxy-10-undecenoic acid
13	5.00	185.0006	186.0079	C_8_H_7_ClO_3_	C_8_H_7_ClO_3_	Chloroatranol
14	5.10	213.1130	214.1203	C_11_H_18_O_4_	C_11_H_18_O_4_	2-Undecenedioic acid
15	5.21	169.0863	170.0936	C_9_H_14_O_3_	C_9_H_14_O_3_	4-Hydroxy-2,5-dimethylcyclohex-1-ene-1-carboxylic acid
16	5.49	163.0392	164.0470	C_9_H_8_O_3_	C_9_H_8_O_3_	*p*-Coumaric acid
17	5.60	209.1181	210.1253	C_12_H_18_O_3_	C_12_H_18_O_3_	6-(1-Oxopentyl)-1-cyclohexene-1-carboxylic acid
18	5.87	215.1286	216.1359	C_11_H_20_O_4_	C_11_H_20_O_4_	Undecanedioic acid
19	6.57	227.1288	228.1360	C_12_H_20_O_4_	C_12_H_20_O_4_	*trans*-Dodec-2-enedioic acid
20	7.44	282.2078	283.2150	C_16_H_29_NO_3_	C_16_H_29_NO_3_	N-Dodecanoyl-L-Homoserine lactone
21	7.67	329.2336	330.2252	-	C_18_H_34_O_5_	9,10-Dihydroxy-8-oxo-12-octadecenoic acid
22	7.77	209.0817	210.0889	C_11_H_14_O_4_	C_11_H_14_O_4_	3,4-Dimethoxyhydrocinnamic acid
23	8.70	373.1294	374.1366	C_20_H_22_O_7_	C_20_H_22_O_7_	4-O-demethyldivaricatic acid
24	9.01	253.1809	254.1882	C_15_H_26_O_3_	C_15_H_26_O_3_	9-Hydroxy-10,12-pentadecadienoic acid
25	9.53	243.1601	244.1675	C_13_H_24_O_4_	C_13_H_24_O_4_	Tridecanedioic acid
26	10.13	359.1139	360.1211	C_19_H_20_O_7_	-	Barbatic acid
27	10.30	375.1086	376.1159	C_19_H_20_O_8_	C_19_H_20_O_8_	8-Hydroxybarbatic acid
28	10.35	373.0931	374.1001	C_19_H_18_O_8_	C_19_H_18_O_8_	Baeomycesic acid
29	10.59	311.2230	312.2302	C_18_H_32_O_4_	C_18_H_32_O_4_	9*Z*-Octadecenedioic acid
30	10.77	351.2178	352.2256	C_20_H_32_O_5_	C_20_H_32_O_5_	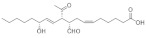 structure proposed
31	10.87	233.1547	234.1619	C_15_H_22_O_2_	C_15_H_22_O_2_	Fukinanolide
32	10.89	323.2230	324.2303	C_19_H_32_O_4_	C_19_H_32_O_4_	*allo*-Protolichestrinic acid
33	10.91	411.0045	412.0118	C_18_H_14_Cl_2_O_7_	C_18_H_14_Cl_2_O_7_	Leoidin
34	10.91	396.9888	397.9961	C_17_H_12_Cl_2_O_7_	C_17_H_12_Cl_2_O_7_	Lecideoidin
35	11.49	403.1399	404.1473	C_21_H_24_O_8_	C_21_H_24_O_8_	4′-*O*-demethylsekikaic acid
36	12.28	313.2388	314.2266	-	C_18_H_34_O_4_	Octadecanedioic acid
37	13.00	269.2124	270.2195	C_16_H_30_O_3_	C_16_H_30_O_3_	2-Oxopalmitic acid
38	13.04	389.1245	390.1315	C_20_H_22_O_8_	C_20_H_22_O_8_	8-Hydroxydiffractaic acid
39	13.34	293.2124	294.2202	C_18_H_30_O_3_	C_18_H_30_O_3_	2-Hydroxylinolenic acid
40	13.58	291.1968	292.2041	C_18_H_28_O_3_	C_18_H_28_O_3_	*α*-Licanic acid
41	14.38	295.2279	296.2351	C_18_H_32_O_3_	C_18_H_32_O_3_	2-Hydroxylinoleic acid isomer
42	14.39	455.1711	456.1783	C_25_H_28_O_8_	C_25_H_28_O_8_	Glomelliferonic acid
43	14.40	295.2278	296.2352	-	C_18_H_32_O_3_	18-Hydroxylinoleic acid
44	14.68	321.2437	322.2509	C_20_H_34_O_3_	C_20_H_34_O_3_	Hydroxyeicosatrienoic acid
45	14.95	305.2125	306.2197	C_19_H_30_O_3_	C_19_H_30_O_3_	14-Oxo-7,10,12-nonadecatrienoic acid
46	15.19	297.2436	298.2508	C_18_H_34_O_3_	C_18_H_34_O_3_	9-Oxooctadecanoic acid
47	15.25	295.2280	296.2351	C_18_H_32_O_3_	C_18_H_32_O_3_	Coriolic acid
48	15.36	297.2435	298.2508	C_18_H_34_O_3_	C_18_H_34_O_3_	Ricinoleic acid
49	15.42	373.0925	374.0999	C_19_H_18_O_8_	C_19_H_18_O_8_	**Atranorin**
50	15.43	177.0187	178.0259	C_9_H_6_O_4_	C_9_H_6_O_4_	6,7-Dihydroxycoumarin
51	15.44	277.2536	278.2610	C_19_H_34_O	C_19_H_34_O	7,10,12-nonadecatrien-1-ol
52	15.46	365.2330	366.2403	C_21_H_34_O_5_	C_21_H_34_O_5_	Muronic acid
53	15.70	277.2538	278.2612	C_18_H_30_O_2_	C_18_H_30_O_2_	Linolenic acid
54	15.98	367.2488	368.2562	C_21_H_36_O_5_	C_21_H_36_O_5_	Murolic acid
55	16.16	311.2594	312.2667	C_19_H_36_O_3_	C_19_H_36_O_3_	Lichesterylic acid
56	16.29	461.2550	462.2619	C_26_H_38_O_7_	C_26_H_38_O_7_	Unidentified
57	16.37	471.3481	472.3553	C_30_H_48_O_4_	C_30_H_48_O_4_	Unidentified
58	16.45	381.2282	382.2356	C_21_H_34_O_6_	C_21_H_34_O_6_	Praesorediosic acid
59	16.47	210.9834	211.9873	C_9_H_5_ClO_4_	C_9_H_5_ClO_4_	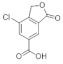 7-Chloro-3-oxo-1,3-dihydroisobenzofuran-5- carboxylic acid.
60	16.47	381.2285	382.2355	C_21_H_34_O_6_	C_21_H_34_O_6_	19-Acetoxy-protolichesterinic acid
61	16.51	407.0540	408.0611	C_19_H_17_ClO_8_	C_19_H_17_ClO_8_	**Chloroatranorin**
62	16.90	421.9285	422.9363	C_16_H_10_Cl_4_O_5_	C_16_H_10_Cl_4_O_5_	Diploicin
63	16.97	201.1493	202.1564	C_11_H_22_O_3_	C_11_H_22_O_3_	11-Hydroxyundecanoic acid
64	16.97	385.2960	386.3032	C_22_H_42_O_5_	C_22_H_42_O_5_	Unidentified
65	17.59	387.2544	388.2616	C_24_H_36_O_4_	C_24_H_36_O_4_	Unidentified
66	17.68	389.1242	390.1314	C_20_H_22_O_8_	C_20_H_22_O_8_	Leprolomin
67	17.69	395.2442	396.2520	C_22_H_36_O_6_	-	structure proposed 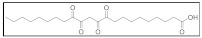
68	17.80	357.0983	358.1057	C_19_H_18_O_7_	C_19_H_18_O_7_	structure proposed: 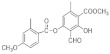
69	17.97	449.3276	450.3354	C_35_H_38_O_7_	C_35_H_38_O_7_	Unidentified
70	18.04	253.2173	254.2244	C_16_H_30_O_2_	C_16_H_30_O_2_	Palmitoleic acid
71	18.33	241.2172	242.2245	C_15_H_30_O_2_	C_15_H_30_O_2_	Pentadecanoic acid
72	18.35	455.3531	456.3605	C_30_H_48_O_3_	C_30_H_48_O_3_	Ursolic acid
73	18.50	279.2330	280.2403	C_18_H_32_O_2_	C_18_H_32_O_2_	Linoleic acid
74	18.66	299.2595	300.2667	C_18_H_36_O_3_	C_18_H_36_O_3_	2-Hydroxyoctadecanoic acid
75	18.78	279.2332	280.2403	C_18_H_32_O_2_	C_18_H_32_O_2_	Linoleic acid isomer, *cis,trans*
76	19.03	497.2548	498.2626	C_29_H_38_O_7_	-	Superpicrolichenic acid
77	19.16	255.2329	256.2401	C_16_H_32_O_2_	C_16_H_32_O_2_	Palmitic acid
78	19.49	281.2485	282.2559	C_18_H_34_O_2_	C_18_H_34_O_2_	Oleic acid
79	19.62	269.2488	270.2561	C_17_H_34_O_2_	C_17_H_34_O_2_	15-Methylhexadecanoic acid
80	19.89	269.2488	270.2561	C_17_H_34_O_2_	C_17_H_34_O_2_	Heptadecanoic acid
81	19.99	483.3481	484.3553	C_31_H_48_O_4_	C_31_H_48_O_4_	Unidentified
82	20.07	327.2543	328.2616	C_20_H_40_O_3_	C_20_H_40_O_3_	2-Hydroxyeicosanoic acid
83	20.17	473.2548	474.2626	C_27_H_38_O_7_	C_27_H_38_O_7_	Unidentified
84	20.54	283.2643	284.2716	C_18_H_36_O_2_	C_18_H_36_O_2_	Stearic acid (octadecanoic acid)
85	20.74	309.2801	310.2875	C_20_H_38_O_2_	C_20_H_38_O_2_	Eicosenoic acid
86	21.02	441.3377	442.3448	C_29_H_46_O_3_	C_29_H_46_O_3_	Unidentified
87	21.14	297.2801	298.2873	C_19_H_38_O_2_	C_19_H_38_O_2_	Nonadecanoic acid
88	21.70	311.2957	312.3029	C_20_H_40_O_2_	C_20_H_40_O_2_	(Eicosanoic acid) Arachidic acid
89	22.02	597.4164	598.4242	C_37_H_58_O_6_	-	Unidentified
90	22.02	669.4739	670.4817	C_41_ H_66_ O_7_	-	Unidentified
91	22.02	545.2758	546.2836	C_30_H_42_O_9_	C_30_H_42_O_9_	Structure proposed and confirmed by the ions 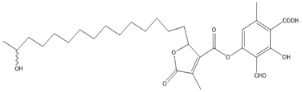
92	22.52	579.2368	580.2446	C_36_H_36_O_7_	-	Unidentified
93	22.66	753.4057	754.4135	C_37_H_62_O_14_	C_37_H_62_O_14_	Unidentified
94	22.68	637.4841	638.4908	C_41_H_66_O_5_	C_41_H_66_O_5_	Unidentified
95	22.69	751.4779	752.4857	C_45_H_38_O_9_	C_45_H_38_O_9_	Unidentified
96	23.78	603.3334	604.3412	C_37_H_48_O_7_	C_37_H_48_O_7_	Unidentified
97	23.81	633.3798	634.3876	C_39_H_54_O_7_	-	Unidentified
98	24.12	467.4109	468.4187	C_30_H_56_O_4_	-	12-Triacontenedioic acid
99	24.51	605.3483	606.3561	C_37_H_50_O_7_	C_37_H_50_O_7_	Unidentified

**Table 6 molecules-27-05229-t006:** Samples *Parmotrema hypoleucinum* (in *Olea europea*), Hexane extract part insoluble in MeOH, **1-Cires** and *Parmotrema hypoleucinum* (in *Quercus coccifera*), Hexane extract part insoluble in MeOH, **2-Cires**.

Nº	RT	[M-H]^−^	MW Calc	Formula	Formula	Compounds
				**1-Cires**	**2-Cires**	
1	2.67	146.9397	147.9475	C_4_H_4_O_6_	-	Dihydroxyfumaric acid
2	0.04	190.9281	191.9359	C_5_H_4_O_8_	-	Methanetetracarboxylic acid
3	1.80	116.9276	117.9354	C_4_H_6_O_4_	-	Butendioic acid
4	2.19	112.9845	113.9932	C_4_H_2_O_4_	C_4_H_2_O_4_	2,3-Dioxobuten-1,4-dial
5	2.72	187.0971	188.1049	C_9_H_16_O_4_	C_9_H_16_O_4_	3,5-Dimethoxyciclohexanecarboxilic acid
6	4.12	201.1129	202.1207	C_10_H_18_O_4_	C_10_H_18_O_4_	2,4-Dihydroxy-3,5,6-trimethylcyclohexane-1-carboxylic acid
7	5.51	163.0395	164.0473	C_9_H_8_O_3_	-	*p*-Coumaric acid
8	5.85	268.1919	269.1997	C_12_H_28_O_6_	C_12_H_28_O_6_	Unidentified
9	6.57	227.1286	228.1364	-	C_12_H_20_O_4_	*trans*-Dodec-2-enedioic acid
10	7.46	282.2077	283.2155	C_16_H_29_NO_3_	C_16_H_29_NO_3_	*N*-Dodecanoyl-*L*-homoserine lactone
11	8.09	174.9556	175.9634	C_5_H_4_O_7_	-	2-Hydroxy-3,4-dioxopentanedioc acid
12	8.15	293.1762	294.1840	C_17_H_26_O_4_	-	gingerol
13	8.39	323.2230	324.2308	-	C_19_H_32_O_4_	allo-protolichestrinic acid
14	9.50	350.2337	351.2415	C_17_H_34_O_7_	C_17_H_34_O_7_	Xylitollaurate
15	10.88	233.1546	234.1624	-	C_15_H_22_O_2_	Fukinanolide
16	11.43	334.2389	335.2467	C_17_H_34_O_6_	C_17_H_34_O_6_	Unidentified
17	11.75	311.2231	312.2309	C_18_H_32_O_4_	C_18_H_32_O_4_	9Z-octadecenedioic acid
18	13.16	293.2126	294.2204	-	C_18_H_30_O_3_	2-Hydroxylinolenic acid
19	13.95	319.2280	320.2358	-	C_20_H_32_O_3_	5-Hydroxyeicosatetraenoic acid
20	14.37	295.2280	296.2358	C_18_H_32_O_3_	C_18_H_32_O_3_	18-Hydroxylinoleic acid
21	14.67	321.2436	322.2514	C_20_H_34_O_3_	-	Hydroxyeicosatrienoic acid
22	14.90	346.2390	347.2468	C_18_H_34_O_6_	C_18_H_34_O_6_	9,10,14-trihydroxy-12-oxooctadecanoic acid
23	14.98	297.2434	298.2512	C_18_H_34_O_3_	C_18_H_34_O_3_	9-oxooctadecanoic acid
24	15.36	177.0186	178.0264	C_9_H_6_O_4_	C_9_H_6_O_4_	6,7-Dihydroxycoumarin
25	15.44	373.0929	374.1007	C_19_H_18_O_8_	C_19_H_18_O_8_	**Atranorin**
26	16.42	210.9801	211.9879	C_9_H_5_O_4_Cl	C_9_H_5_O_4_Cl	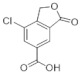 7-chloro-3-oxo-1,3-dihydroisobenzofuran-5- carboxylic acid.
27	16.47	407.0539	408.0617	C_19_H_17_O_8_Cl	C_19_H_17_O_8_Cl	**Chloroatranorin**
28	17.57	277.2175	278.2203	-	C_18_H_30_O_2_	Octadeca-9,12,15-trienoic acid
29	17.65	389.1246	390.1324	C_20_H_22_O_8_	-	8-Hydroxydiffractaic acid,
30	17.77	265.1480	266.1558	C_15_H_22_O_4_	C_15_H_22_O_4_	(4*E*,6*E*,9*E*)-Pentadeca-4,6,9-trienedioic acid
31	18.03	253.2331	254.2249	C_16_H_30_O_2_	C_16_H_30_O_2_	Palmitoleic acid
32	18.28	402.3016	403.3094	-	C_22_H_42_O_6_	Unidentified
33	18.34	241.2173	242.2251	C_15_H_30_O_2_	C_15_H_30_O_2_	Pentadecanoic acid
34	18.35	455.3534	456.3612	C_30_H_48_O_3_	-	Oleanolic acid
35	18.50	279.0936	280.2409	C_18_H_32_O_2_	C_18_H_32_O_2_	Linoleic acid
36	18.66	489.3375	490.3453	-	C_26_H_50_O_8_	Unidentified
37	18.86	403.2645	404.2723	-	C_28_H_36_O_2_	Unidentified
38	18.91	267.2331	268.2409	C_17_H_32_O_2_	C_17_H_32_O_2_	*cis*-9-Heptadecenoic acid
39	19.15	255.2329	256.2407	C_16_H_32_O_2_	C_16_H_32_O_2_	Palmitic acid
40	19.35	459.3271	460.3349	C_25_H_48_O_7_	C_25_H_48_O_7_	Unidentified
41	19.51	281.2487	282.2565	C_18_H_34_O_2_	C_18_H_34_O_2_	Oleic acid
42	19.75	459.3272	460.3350	C_25_H_48_O_7_	-	Methyl glucose isostearate
43	19.89	269.2488	270.2566	C_17_H_34_O_2_	C_17_H_34_O_2_	Heptadecanoic acid
44	19.99	307.2645	308.2723	C_20_H_36_O_2_	C_20_H_36_O_2_	11,14-Eicosadienoic acid
45	20.07	457.3722	458.3800	C_27_H_54_O_5_	C_27_H_54_O_5_	Unidentified
46	20.18	295.2645	296.2723	C_19_H_36_O_2_	C_19_H_36_O_2_	10*E*-nonadecenoic acid
47	20.54	283.2643	284.2721	C_18_H_36_O_2_	C_18_H_36_O_2_	Stearic acid (Octadecanoic acid)
48	20.82	309.2800	310.2878	C_20_H_38_O_2_	C_20_H_38_O_2_	Eicosenoic acid
49	20.86	505.3326	506.3404	-	C_26_H_50_O_9_	Unidentified
50	21.05	457.3722	458.3800	C_27_H_54_O_5_	-	Unidentified
51	21.71	311.2957	312.3035	-	C_20_H_40_O_2_	(Eicosanoic acid) arachidic acid
52	22.62	297.1532	298.1610	C_12_H_26_O_8_	C_12_H_26_O_8_	Unidentified
53	22.67	637.4836	638.4914	C_24_H_60_O_12_N_7_	-	Unidentified
54	22.77	339.3268	340.3346	C_22_H_44_O_2_	C_22_H_44_O_2_	Docosanoic acid
55	23.08	309.1743	310.1821	C_17_H_26_O_5_	C_17_H_26_O_5_	Portentol
56	23.58	353.2003	354.2081	C_19_H_30_O_6_	C_19_H_30_O_6_	Unidentified
57	23.81	311.1689	312.1767	C_13_H_30_O_8_	C_13_H_30_O_8_	Unidentified
58	23.89	367.3579	368.3657	C_24_H_48_O_2_	C_24_H_48_O_2_	Lignoceric acid
59	24.01	397.2266	398.2344	C_21_H_34_O_7_	C_21_H_34_O_7_	Stephanol
60	24.60	293.1793	294.1871	C_17_H_26_O_4_	C_17_H_26_O_4_	Heptadecatrienedioic acid
61	24.94	325.1844	326.1922	C_14_H_30_O_8_	C_14_H_30_O_8_	Unidentified
62	25.33	395.3895	396.3973	C_26_H_52_O_2_	-	Hexacosanoic acid or cerotic acid
63	25.89	337.2055	338.2133	C_19_H_30_O_5_	C_19_H_30_O_5_	6-Oxononadeca-8,11-dienedioic acid
64	26.10	339.2000	340.2078	C_15_H_32_O_8_	C_15_H_32_O_8_	Unidentified
65	26.24	381.2317	382.2395	C_21_H_34_O_6_	C_21_H_34_O_6_	19-Acetoxylichesterinic acid
66	26.75	425.2581	426.2659	C_23_H_38_O_7_	-	Asebotoxin I
67	27.53	321.2106	322.2184	C_19_H_30_O_4_	C_19_H_30_O_4_	Nonadecatrienedioic Acid
68	27.85	304.9143	305.9221	Noformula	-	Unidentified

**Table 7 molecules-27-05229-t007:** Samples *Parmotrema hypoleucinum* (in *Olea europea*), Chloroform extract, **1-Chloroform** and *Parmotrema hypoleucinum* (in *Quercus coccifera*), Chloroform extract, **2-Chloroform**.

Nº	RT	[M-H]^−^	Mass Calc	Formula	Formula	Compounds
				1-Chloroform	2-Chloroform	
1	0.98	174.9557	175.9635	C_5_H_4_O_7_	-	2-Hydroxy-3,4-dioxopentanedioic acid
2	1.01	145.0975	146.1053	-	C_7_H_14_O_3_	2-Hydroxyheptanoic acid
3	1.05	112.9845	113.9923	-	C_4_H_2_O_4_	Acetylenedicarboxylic acid (Squaric acid)
4	1.12	182.9882	183.9960	-	C_7_H_4_O_6_	Chelidonic acid
5	1.14	341.1091	342.1169	-	C_12_H_22_O_11_	Sucrose
6	1.52	215.0097	216.0175	C_8_H_8_O_7_	C_8_H_8_O_7_	D-diacetyltartaric anhydride
7	1.74	433.0778	434.0856	C_20_H_18_O_11_	C_20_H_18_O_11_	Avicularin
8	1.76	403.0675	404.0753	C_19_H_16_O_10_	C_19_H_16_O_10_	Euxanthic acid
9	1.92	417.0468	418.0546	C_19_H_14_O_11_	-	Shoyuflavone C
10	2.41	433.0780	434.0858	C_20_H_18_O_11_	C_20_H_18_O_11_	Morin 3-alpha-L-arabinopyranoside
11	2.57	401.0518	402.0596	C_19_H_14_O_10_	C_19_H_14_O_10_	Shoyuflavone B
12	2.64	403.0672	404.0750	C_19_H_16_O_10_	C_19_H_16_O_10_	Euroxanthone B
13	3.15	447.0934	448.1012	C_21_H_20_O_11_	C_21_H_20_O_11_	Quercitrin
14	3.23	401.0515	402.0593	C_19_H_14_O_10_	C_19_H_14_O_10_	**Constictic acid**
15	3.38	182.9882	183.9960	C_8_H_5_O_3_Cl	C_8_H_5_O_3_Cl	3-Chloro-4-formylbenzoic acid
16	3.48	187.0972	188.1050	C_9_H_16_O_4_	-	Azelaic acid
17	3.63	371.0412	372.0490	C_18_H_12_O_9_	-	**Norstictic acid**
18	3.90	313.0722	314.0800	-	C_17_H_14_O_6_	Cirsimaritin
19	3.91	442.1145	443.1223	C_18_H_29_O_8_Cl_2_	-	Unidentified
20	3.99	373.0567	374.0645	-	C_18_H_14_O_9_	Menegazziaic acid
21	4.02	399.0361	400.0439	C_19_H_12_O_10_	-	Kynapcin-28
22	4.13	357.0617	358.0695	C_18_H_14_O_8_	C_18_H_14_O_8_	Succinyldisalicylic acid
23	4.24	401.0515	402.0593	C_19_H_14_O_10_	-	Siphulellic acid
24	4.27	373.0568	374.0646	C_18_H_14_O_9_	-	Protocetraric acid
25	5.03	417.0830	418.0908	C_20_H_18_O_10_	C_20_H_18_O_10_	Conphysodalic acid
26	5.09	385.0568	386.0646	C_19_H_14_O_9_	C_19_H_14_O_9_	**Stictic acid**
27	5.18	387.0721	388.0799	C_19_H_16_O_9_	C_19_H_16_O_9_	Cryptostictic acid
28	6.02	385.0565	386.0643	C_19_H_14_O_9_	C_19_H_14_O_9_	3,3′-Carbonylbis [6-(methoxycarbonyl)-benzoic acid]
29	6.14	431.0984	432.1062	C_21_H_20_O_10_	C_21_H_20_O_10_	Genistein 7-glucoside (Genistin)
30	6.32	209.0849	210.0927	-	C_11_H_14_O_4_	Sinapyl alcohol
31	7.63	309.1017	310.1095	-	C_15_H_18_O_7_	1-O-cis-cinnamoyl-β-D-glucopyranose
32	8.11	373.0568	374.0646	-	C_18_H_14_O_9_	Menegazziaic acid isomer
33	8.67	371.0408	372.0486	C_18_H_12_O_9_	C_18_H_12_O_9_	**Substictic acid**
34	9.54	293.1763	294.1841	C_17_H_26_O_4_	C_17_H_26_O_4_	Nordihydrocapsiate
35	10.26	771.1205	772.1283	C_38_H_28_O_18_	C_38_H_28_O_18_	fucofuroeckol A hepta-acetate
36	10.61	426.9681	427.9759	-	C_16_H_12_O_14_	Unidentified
37	11.11	475.3278	476.3356	C_25_H_48_O_8_	C_25_H_48_O_8_	Tetrahydroxypentacosanedioic acid
38	12.21	345.0982	346.1060	C_18_H_18_O_7_	-	Isooptusatic acid (or 3′-Methylevernic acid)
39	13.12	265.1482	266.1560	C_15_H_22_O_4_	C_15_H_22_O_4_	EthyI 4-O-methylolivetolcarboxylate
40	13.34	467.0985	468.1063	C_24_H_20_O_10_	C_24_H_20_O_10_	Gyrophoric acid
41	13.36	317.0670	318.0748	C_16_H_14_O_7_	-	Lecanoric acid
42	13.80	503.3593	504.3671	C_27_H_52_O_8_	C_27_H_52_O_8_	Tetraglyceryl monooleate
43	14.47	517.3745	518.3823	C_28_H_54_O_8_	-	13-beta-D-glucosyloxy)docosanoic acid
44	14.83	359.0776	360.0854	-	C_18_H_16_O_8_	Ramalinaic acid
45	15.40	365.2334	366.2412	C_21_H_34_O_5_	C_21_H_34_O_5_	Muronic acid
46	15.87	265.1479	266.1557	C_15_H_22_O_4_	C_15_H_22_O_4_	Ivambrin
47	16.02	367.2491	368.2569	-	C_21_H_36_O_5_	Constipatic acid or Protoconstipatic acid
48	16.28	297.1532	298.1610	C_12_H_26_O_8_	C_12_H_26_O_8_	Unidentified
49	17.14	177.0187	178.0265	C_9_H_6_O_4_	C_9_H_6_O_4_	6,7-dihydroxycoumarin
50	17.37	309.1743	310.1821	C_17_H_26_O_5_	C_17_H_26_O_5_	Portentol
51	18.01	311.1690	312.1768	C_13_H_28_O_8_	C_13_H_28_O_8_	heptahydroxytridecanol
52	18.03	407.0541	408.0619	C_19_H_17_O_8_Cl	C_19_H_17_O_8_Cl	**Chloroatranorin**
53	18.05	210.9801	211.9879	C_7_H_10_O_3_Cl_2_	C_7_H_10_O_3_Cl_2_	2-Methoxy-3,4-dichloro-6-methyltetrahydropyran-5-one
54	18.32	353.2004	354.2082	C_19_H_30_O_6_	C_19_H_30_O_6_	tetraoxononadecanoic acid
55	18.36	421.2265	422.2343	-	C_23_H_34_O_7_	Sarmentologenin
56	18.85	397.2268	398.2346	C_21_H_34_O_7_	-	Stephanol
57	19.13	387.2544	388.2622	-	C_24_H_36_O_4_	Dehydrodeoxycholic acid
58	19.16	441.2530	442.2608	C_23_H_38_O_8_	-	Asebotoxin IV
59	19.32	325.1846	326.1924	C_21_H_26_O_3_	-	Linderanolide
60	19.55	253.2172	254.2250	C_16_H_30_O_2_	C_16_H_30_O_2_	palmitoleic acid (9-cis-hexadecenoic acid)
61	19.64	255.2331	256.2409	C_16_H_32_O_2_	-	palmitic acid
62	19.74	293.1796	294.1874	-	C_17_H_26_O_4_	Gingerol
63	19.92	279.2332	280.2410	-	C_18_H_32_O_2_	Linoleic acid
64	20.35	267.2332	268.241	C_17_H_32_O_2_	C_17_H_32_O_2_	2-Heptadecenoic acid
65	20.51	283.2645	284.2723	C_18_H_36_O_2_	C_18_H_36_O_2_	Stearic acid
66	20.88	281.2488	282.2566	C_18_H_34_O_2_	C_18_H_34_O_2_	Oleic acid
67	22.25	565.3784	566.3862	C_32_H_54_O_8_	-	Unidentified
68	23.08	679.4650	680.4728	C_35_H_68_O_12_	C_35_H_68_O_12_	Unidentified
69	23.11	761.5977	762.6055	C_35_H_68_O_12_	C_35_H_68_O_12_	Unidentified
70	23.14	395.3897	396.3975	-	C_26_H_52_O_2_	Hexacosanoic acid
71	23.84	337.2057	338.2135	C_19_H_30_O_5_	C_19_H_30_O_5_	Idebenone
72	23.91	367.3583	368.3661	C_24_H_48_O_2_	-	Lignoceric acid (tetracosanoic acid)
73	24.13	637.4844	638.4922	-	C_34_H_70_O_10_	Unidentified
74	24.25	639.3973	640.4051	C_31_H_60_O_13_	-	Unidentified
75	24.44	381.3741	382.3819	C_25_H_50_O_2_	-	Pentacosanoic acid
76	24.56	339.2000	340.2078	C_15_H_32_O_8_	C_15_H_32_O_8_	Heptahydroxypentadecanol
77	24.60	679.4650	680.4728	C_35_H_68_O_12_	C_35_H_68_O_12_	Unidentified
78	24.80	535.3132	536.3210	C_26_H_48_O_11_	C_26_H_48_O_11_	Unidentified

**Table 8 molecules-27-05229-t008:** Samples *Parmotrema hypoleucinum* (in *Olea europea*), Ethanol extract, **1-Ethanol** and *Parmotrema hypoleucinum* (in *Quercus coccifera*), Ethanol extract, **2-Ethanol**.

Nº	RT	[M-H]^−^	MW Calc	Formula	Formula	Compounds
				1-Ethanol	2-Ethanol	
1	0.98	174.9556	175.9634	C_5_H_4_O_7_	C_5_H_4_O_7_	2-Hydroxy-3,4-dioxopentanedioic acid
2	1.05	112.9845	113.9923	C_4_H_2_O_4_	C_4_H_2_O_4_	Acetylenedicarboxylic acid (Squaric acid)
3	1.15	311.1156	312.1234	C_15_H_20_O_7_	C_15_H_20_O_7_	Neoanisatin
4	1.19	151.0603	152.0681	C_5_H_12_O_5_	C_5_H_12_O_5_	Arabitol
5	1.82	182.9882	183.9960	C_7_H_4_O_6_	C_7_H_4_O_6_	Chelidonic acid
6	2.99	401.0517	402.0595	C_19_H_14_O_10_	C_19_H_14_O_10_	**Constictic acid**
7	3.95	357.0617	358.0695	C_18_H_14_O_8_	C_18_H_14_O_8_	Hyposalazinic acid, Psoromic acid or Virensic acid
8	4.17	519.1147	520.1225	C_24_H_24_O_13_	-	Eujambolin
9	4.91	387.0722	388.0800	C_19_H_16_O_9_	C_19_H_16_O_9_	Cryptostictic acid
10	5.06	385.0568	386.0646	C_19_H_14_O_9_	C_19_H_14_O_9_	**Stictic acid**
11	5.19	417.0830	418.0908	C_20_H_18_O_10_	C_20_H_18_O_10_	Juglanin
12	5.71	373.0566	374.0644	C_18_H_14_O_9_	C_18_H_14_O_9_	Protocetraric acid
13	5.88	431.0985	432.1063	C_21_H_20_O_10_	-	Genistin
14	6.51	328.0597	329.0675	C_20_H_11_O_4_N	-	Unidentified
15	6.80	163.0393	164.0471	C_9_H_8_O_3_	C_9_H_8_O_3_	Coumaric acid
16	7.25	359.0775	360.0853	-	C_18_H_16_O_8_	Conhypoprotocetraric acid, [40]
17	7.83	399.0723	400.0801	C_20_H_16_O_9_	-	**Methylstictic acid**
18	8.44	371.0408	372.0486	C_18_H_12_O_9_	C_18_H_12_O_9_	**Substictic acid**
19	8.98	209.0453	210.0531	C_10_H_10_O_5_	-	5,6-Dihydroxy-7-methoxy-4-methyl-2-benzofuran-1(3H)-one
20	9.24	293.1763	294.1841	C_17_H_26_O_4_	C_17_H_26_O_4_	(+)-[6]-Gingerol
21	9.33	413.0881	414.0959	C_21_H_18_O_9_	C_21_H_18_O_9_	Vesuvianic acid
22	9.84	461.3123	462.3201	-	C_24_H_46_O_8_	Unidentified
23	10.28	389.2911	390.2989	-	C_21_H_42_O_6_	9,10,12,13-tetrahydroxyheneicosanoic acid
24	10.49	243.0065	244.0143	C_10_H_9_O_5_Cl	-	(4-Chloro-2-formyl-6-methoxyphenoxy)acetic Acid
25	10.97	475.3276	476.3354	C_25_H_48_O_8_	C_25_H_48_O_8_	Tetrahydroxypentacosanedioic acid
26	11.38	265.1481	266.1559	C_15_H_22_O_4_	C_15_H_22_O_4_	EthyI 4-O-methylolivetolcarboxylate
27	11.42	403.3068	404.3146	C_22_H_44_O_6_	C_22_H_44_O_6_	9,10,12,13-Tetrahydroxydocosanoic acid
28	11.62	447.3331	448.3409	-	C_24_H_48_O_7_	D-Glucitol monostearate
29	11.96	489.3434	490.3512	C_26_H_50_O_8_	C_26_H_50_O_8_	Icosanedioic acid bis(2,3-dihydroxypropyl) ester
30	12.05	345.0982	346.1060	C_18_H_18_O_7_	C_18_H_18_O_7_	Isooptusatic acid (or 3′-Methylevernic acid)
31	12.54	417.3224	418.33.02	C_23_H_46_O_6_	C_23_H_46_O_6_	Heptadecyl D-glucoside
32	13.20	343.0823	344.0901	C_18_H_16_O_7_	C_18_H_16_O_7_	Usnic acid
33	13.57	431.3379	432.3457	C_24_H_48_O_6_	C_24_H_48_O_6_	6-Ethyl-6-n-pentyl-pentadecan -4,5,7,8,15-pentol-I5-acetate
34	13.73	503.3593	504.3671	C_27_H_52_O_8_	C_27_H_52_O_8_	Tetraglyceryl monooleate
35	14.25	309.1746	310.1824	C_17_H_26_O_5_	C_17_H_26_O_5_	Portentol
36	14.38	293.2126	294.2204	C_18_H_30_O_3_	C_18_H_30_O_3_	17-Hydroxylinolenic acid
37	14.40	517.3750	518.3828	C_28_H_54_O_8_	C_28_H_54_O_8_	13-(beta-D-Glucosyloxy)docosanoic acid
38	15.34	365.2335	366.2413	C_21_H_34_O_5_	C_21_H_34_O_5_	Muronic acid
39	16.68	297.2438	298.2516	C_18_H_34_O_3_	C_18_H_34_O_3_	Ricinoleic acid
40	18.00	311.1691	312.1769	C_13_H_28_O_8_	C_13_H_28_O_8_	Heptahydroxytridecanol
41	18.20	353.2005	354.2083	C_19_H_30_O_6_	C_19_H_30_O_6_	Tetraoxononadecanoic acid
42	18.71	397.2267	398.2345	C_21_H_34_O_7_	C_21_H_34_O_7_	Stephanol
43	18.93	421.2269	422.2347	C_23_H_34_O_7_	C_23_H_34_O_7_	Sarmentologenin
44	19.09	441.2531	442.2609	C_23_H_38_O_8_	C_23_H_38_O_8_	Asebotoxin IV
45	19.41	485.2796	486.2874	C_25_H_42_O_9_	C_25_H_42_O_9_	2-(7Z,10Z,13Z)-hexadecatrienoyl-3-(β-D-galactosyl)-sn-glycerol
46	19.54	253.2173	254.2251	C_16_H_30_O_2_	C_16_H_30_O_2_	palmitoleic acid (9-cis-hexadecenoic acid)
47	19.83	241.2172	242.2250	-	C_15_H_30_O_2_	Pentadecanoic acid
48	19.93	279.2332	280.2410	C_18_H_32_O_2_	C_18_H_32_O_2_	Linoleic acid
49	20.03	293.1795	294.1873	C_17_H_26_O_4_	C_17_H_26_O_4_	Gingerol
50	20.50	325.1844	326.1922	C_21_H_26_O_3_	C_21_H_26_O_3_	Linderanolide
51	20.63	255.2329	256.2407	C_16_H_32_O_2_	C_16_H_32_O_2_	Palmitic acid
52	20.89	281.2488	282.2566	C_18_H_34_O_2_	C_18_H_34_O_2_	Oleic acid
53	21.18	325.1846	326.1924	-	C_14_H_30_O_8_	Hexahydroxytetradecan-1-ol
54	21.38	591.2616	592.2694	C_34_H_40_O_9_	-	Scortechinone F
55	22.01	283.2645	284.2723	C_18_H_36_O_2_	C_18_H_36_O_2_	Stearic acid
56	22.21	761.5975	762.6053	C_46_H_82_O_8_	C_46_H_82_O_8_	Unidentified
57	22.92	373.0932	374.1010	C_19_H_18_O_8_	-	Baeomycesic acid
58	22.94	535.3136	536.3214	C_26_H_48_O_11_	-	Unidentified
59	23.31	311.2958	312.3036	C_20_H_40_O_2_	C_20_H_40_O_2_	Arachidic acid
60	23.54	337.2057	338.2135	C_19_H_30_O_5_	C_19_H_30_O_5_	Idebenone
61	24.82	339.2000	340.2078	C_15_H_32_O_8_	C_15_H_32_O_8_	Heptahydroxypentadecanol
62	26.80	367.3584	368.3662	C_24_H_48_O_2_	C_24_H_48_O_2_	Tetracosanoic acid
63	27.03	381.2319	382.2397	C_21_H_34_O_6_	C_21_H_34_O_6_	Praesorediosic acid or Protopraesorediosic acid [41,42,43]

**Table 9 molecules-27-05229-t009:** Solvents used, % of solvent **A** and % of solvent **B**.

Time (min)	%A	%B
0	50	50
20	0	100
25	0	100
26	50	50

Total analysis time was set to 30 min.

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

*praesorediosum* (Nyl.) hale, parmeliaceae. Rec. Nat. Prod..

