# Peer review of "Phytochemical Composition of Lichen Parmotrema hypoleucinum (J. Steiner) Hale from Algeria"

_molecules, 2022, doi:10.3390/molecules27165229_

Round 1
Reviewer 1 Report
This is very thorough research on lichen photochemistry. The authors have done great work on this. Especially for the compound identified by GCMS, if we have retention indices and reference RI that will surely help to make this research publication quality. For unidentified peaks, I would like to see a fragmentation pattern.
Author Response
First of all thank you to the referees for their comments that has helped to improve the manuscript.
I send you the revised version with all the changes suggested by the referees done.
Thank you for your help and please tell me if you need anything else
Best regards
Jesús L. Rodilla
Reviewer 2 Report
The manuscript entitled “Phytochemical composition of Lichen Parmotrema hypoleucinum (J. Steiner) Hale from Algeria” addresses a study for the chemical composition of hexane, chloroform and ethanol extracts from two samples of Parmotrema hypoleucinum lichen collected in Algeria.
The manuscript can be considered for publication after its improvement.
My suggestions are:
The abstract lacks purpose and a short conclusion.
Introduction:
The authors do not respect the rules of technical editing of the Journal. Quotations are written only by number. Ex line 68
The purpose is not detailed in the introduction either.
Results and Discussion
The results are well detailed, but the discussions are very brief. I recommend improving them.
The conclusion is very long and difficult to understand.
I recommend reformulating it.
Bibliographic references do not comply with the requirements of the Journal
Author Response

(The authors gave the same response as above.)

Reviewer 3 Report
dear author see the comments in the attached file.
Improve the mentioned sections.
Table must be added in supplementary file

Author Response

(The authors gave the same response as above.)

Round 2
Reviewer 2 Report
The manuscript has been improved.
It can be published in its current form.